# Anti-Ischemic Effect of Leptin in the Isolated Rat Heart Subjected to Global Ischemia-Reperfusion: Role of Cardiac-Specific miRNAs

Ekaterina A. Polyakova *, Evgeny N. Mikhaylov [ID], Sarkis M. Minasian, Mikhail M. Galagudza and Evgeny V. Shlyakhto

Almazov National Medical Research Centre, Institute of Experimental Medicine, 197341 Saint-Petersburg, Russia
* Correspondence: polyakova_ea@yahoo.com

**Abstract:** Background: Leptin is an obesity-associated adipokine that has been implicated in cardiac protection against ischemia-reperfusion injury (IRI). In this study, concentration-dependent effects of leptin on myocardial IRI were investigated in the isolated rat heart. In addition, we analyzed myocardial miRNAs expression in order to investigate their potential involvement in leptin-mediated cardioprotection. Methods: The effect of leptin on IRI was examined in Langendorff-perfused rat hearts preconditioned with two leptin concentrations (1.0 nM and 3.1 nM) for 60 min. The hearts were subjected to 30 min global ischemia and 120 min reperfusion with buffer containing leptin in the respective concentration. Heart function and arrhythmia incidence were analyzed. Infarct size was assessed histochemically. Expression of miRNA-144, -208a, -378, and -499 was analyzed in the ventricular myocardium using RT-PCR. Results: The addition of 1.0 nM leptin to the buffer exerted an infarct-limiting effect, preserved post-ischemic ventricular function, and prevented reperfusion arrhythmia compared to 3.1 nM leptin. Myocardial expression of miRNA-208a was decreased after heart exposure to 1.0 nM leptin and significantly elevated in the hearts perfused with leptin at 3.1 nM. Conclusion: Acute administration of leptin at low dose (1.0 nM) results in cardiac protection against IRI. This effect is associated with reduced myocardial expression of miRNA-208a.

**Keywords:** leptin; heart; ischemia; reperfusion; infarct size; rat; miRNA



## 1. Introduction

Leptin, a protein secreted by adipocytes, plays an important role in body weight regulation through its effects on energy balance and food intake. Leptin is mainly synthesized in white adipose tissue, although the expression levels and secretion rates differ between body fat depots [1]. There is a positive association between body mass index and leptin levels [1–3]. The correlation between fat mass and atherogenesis was confirmed in obese leptin-deficient *ob/ob* mice in a previous study [4]. Moreover, obesity-associated hyperleptinemia is an important biomarker that predicts cardiovascular outcomes, suggesting that leptin plays an important role in obesity-associated cardiovascular disorders [5].

However, the obesity paradox indicates that obesity confers a protective effect in patients with recurrent vascular events and a high risk of mortality despite its deleterious role in disease development, and the obesity paradox is a widely debated phenomenon [6]. Although the evidence for the obesity paradox has been presented for various pathologies, some authors argue that it may be explained by selection bias or unmeasured residual confounding factors [7]. It is also known that miRNAs are involved in the regulation of post-transcriptional gene expression and play important role in cardiac metabolism [8]. However, the regulation of heart-enriched miRNAs in the settings of cardiac ischemia-reperfusion damage is still not fully understood, especially in hyperleptinemia [9].

Importantly, the metabolic effects of leptin depend on an animal's nutritional status; therefore, many leptin-induced changes observed during leptin treatment in fasted animals

are reduced or absent in the fed control group [10,11]. The ability of leptin to lower the body weight of *ob/ob* mice occurs along with leptin-induced improved glycaemic control. Moreover, a low leptin dose improves glycaemic control without decreasing body weight in *ob/ob* mice, and pair-feeding experiments have indicated that improved glycaemic control can be dissociated from leptin-induced decreases in food intake [12].

The chronic administration of high-dose leptin increases mean arterial pressure and heart rate (HR), with the latter possibly involving cardiac leptin receptors [13,14]. Moreover, the heart expresses leptin receptors, synthesizes, and releases leptin into the coronary effluent, thus raising the possibility that the leptin of cardiac origin feeds back on cardiomyocytes to exert physiological effects [15]. Acute leptin administration has been shown to attenuate ischemic injury after stroke via the induction of anti-apoptotic pathways [16,17]. Myocardial protection induced by the leptin administered during reperfusion in the isolated heart protocol is mediated through a mechanism involving JAK/STAT signaling pathway activation coupled with mitochondrial permeability transition pore inhibition [18]. At the same time, based on the understanding of the different roles of microRNAs (miRNA) in coronary artery disease associated with obesity, as well as their potential as therapeutic targets, we have considered that estimation of the expression of miRNA-144, miRNA-208a, miRNA-378, and miRNA-499 in the myocardium after ischemia/reperfusion damage is particularly important. Recent studies have provided increasing evidence that those miRNAs may play a role in cardiac ischemic injury, including arrhythmia, fibrosis, angiogenesis, and apoptosis [8,9]. Nevertheless, our current knowledge about the regulation and function of specific miRNAs in ischemic heart disease is still quite limited.

We hypothesized that the effect of leptin on the heart depends on its concentration. This study aimed to compare the effects of low and high leptin concentrations on ischemia-reperfusion injury (IRI) and miRNA expression in isolated rat heart preparation.

## 2. Results

### 2.1. Left Ventricular Developed Pressure (LVDP) and Left Ventricular End-diastolic Pressure (LVEDP)

The LVDP and LVEDP values were similar among groups during the 60-min conditioning period of the experiment. No spontaneous mechanical activity was observed during the ischemia period. The leptin 3.1 nM (LEP3.1) group showed no cardiac contraction during the initial 5 min of reperfusion. The recovery of LVDP in the LEP3.1 group was significantly greater than that in the control (CON) group ($p < 0.01$; Figure 1). However, the most complete LVDP recovery was observed in the leptin 1.0 nM (LEP1.0) group ($p < 0.0001$ and $p < 0.001$ when compared with the CON and LEP3.1 groups, respectively; Figure 1).

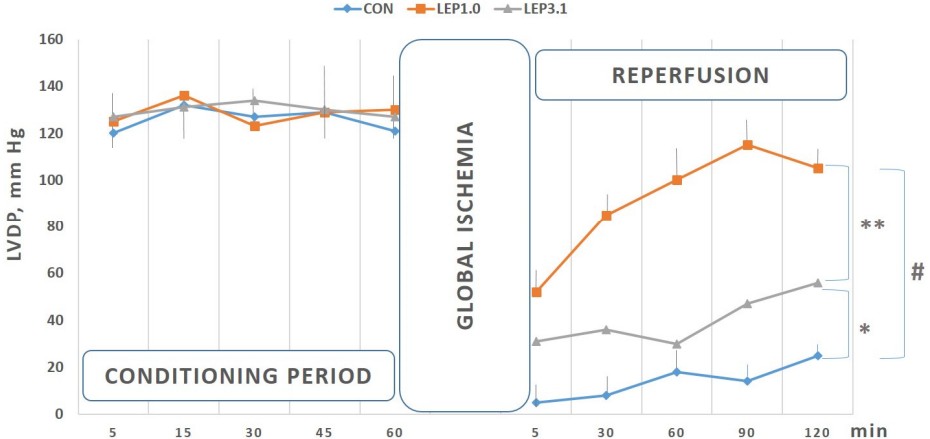

**Figure 1.** Left ventricular developed pressure (LVDP) in the control (CON), leptin 1.0 nM (LEP1.0), and leptin 3.1 nM (LEP3.1) groups. Data are presented as mean $\pm$ standard deviation. * $p < 0.01$; ** $p < 0.001$; # $p < 0.0001$.

The LVEDP after the ischemia period was significantly lower in the LEP3.1 group than in the CON group (*p* < 0.001; Figure 2). The post-ischemic LVEDP remained normal in the LEP1.0 group versus the CON and LEP3.1 groups (*p* < 0.0001 and *p* = 0.021, respectively; Figure 2).

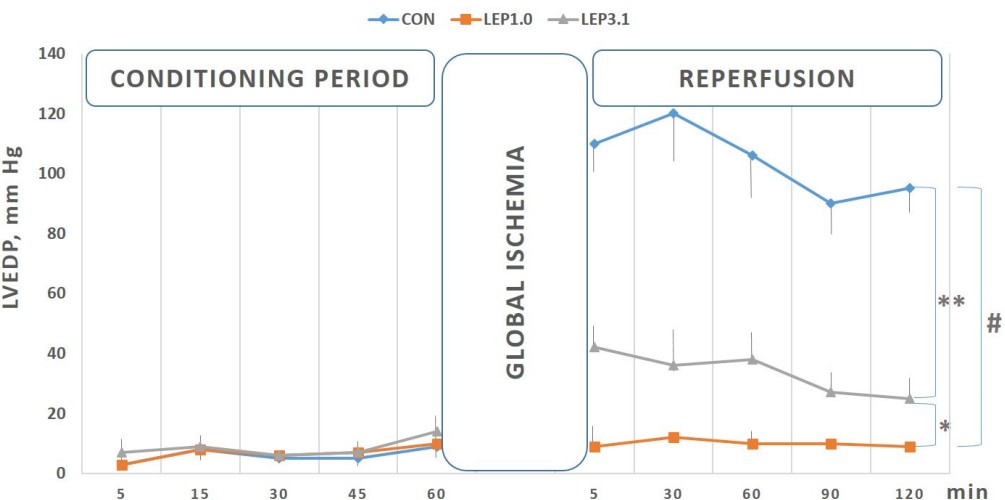

**Figure 2.** Left ventricular end-diastolic pressure (LVEDP) in the control (CON), leptin 1.0 nM (LEP1.0), and leptin 3.1 nM (LEP3.1) groups. Data are presented as mean ± standard deviation. * *p* = 0.021; ** *p* < 0.001; # *p* < 0.0001.

## 2.2. Ischemic Contracture

The dynamics of left ventricle (LV) pressure during global ischemia in the CON, LEP1.0, and LEP3.1 groups is shown in Figure 3. Ischemic contracture was defined as at least a three-fold increase in LV pressure at any time point during the ischemic period relative to the LV pressure after 5 min of ischemia. No cases of ischemic contracture occurred in the LEP1.0 group. The LV pressures during ischemia were significantly higher in the CON and LEP3.1 groups than in the LEP1.0 group (*p* < 0.0001; Figure 3).

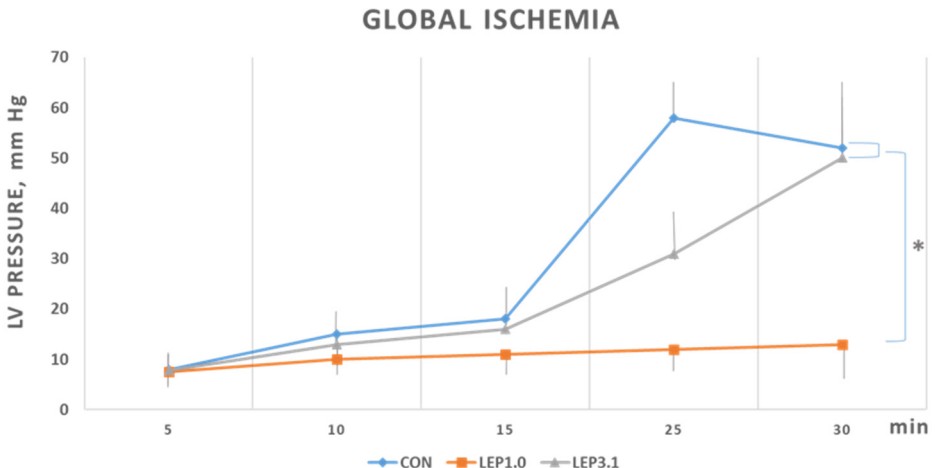

**Figure 3.** Ischemic contracture in the control (CON), leptin 1.0 nM (LEP1.0), and leptin 3.1 nM (LEP3.1) groups. Data are presented as mean ± standard deviation. * *p* < 0.0001.

## 2.3. Coronary Flow Rate (CFR) and Heart Rate (HR)

The mean CFR and HR were similar among the groups during the 60 min conditioning period of the experiment and during the 120 min reperfusion period (Table 1). No significant inter- or intragroup differences in CFR or HR were observed.

**Table 1.** Heart rate and coronary flow rate by experimental group.

| Series | 60 min Conditioning Period, min | | | | | Reperfusion, min | | | | |
|---|---|---|---|---|---|---|---|---|---|---|
| | 5 | 15 | 30 | 45 | 60 | 5 | 30 | 60 | 90 | 120 |
| | Heart rate, beats/min | | | | | | | | | |
| CON | 256 ± 16 | 250 ± 13 | 232 ± 21 | 260 ± 19 | 255 ± 23 | 201 ± 28 | 184 ± 18 | 216 ± 24 | 219 ± 21 | 251 ± 38 |
| LEP1.0 | 263 ± 22 | 258 ± 16 | 246 ± 17 | 257 ± 18 | 264 ± 21 | 183 ± 31 | 199 ± 22 | 234 ± 19 | 229 ± 28 | 247 ± 17 |
| LEP3.1 | 260 ± 19 | 242 ± 23 | 260 ± 14 | 268 ± 20 | 259 ± 17 | - | 192 ± 15 | 212 ± 33 | 227 ± 21 | 230 ± 25 |
| | Coronary flow rate, mL/min | | | | | | | | | |
| CON | 11.6 ± 1.3 | 10.8 ± 0.9 | 11.2 ± 2.4 | 12.1 ± 1.6 | 11.0 ± 1.7 | 3.6 ± 0.5 | 3.8 ± 1.0 | 4.2 ± 1.4 | 3.9 ± 0.7 | 4.2 ± 1.3 |
| LEP1.0 | 11.1 ± 1.0 | 11.7 ± 1.1 | 10.9 ± 1.3 | 12.2 ± 1.9 | 11.5 ± 0.8 | 4.6 ± 1.3 | 4.7 ± 0.9 | 5.3 ± 1.7 | 5.8 ± 1.1 | 5.7 ± 1.5 |
| LEP3.1 | 12.6 ± 2.1 | 12.3 ± 1.8 | 11.6 ± 2.3 | 11.9 ± 2.0 | 10.1 ± 1.1 | 2.7 ± 0.7 | 4.1 ± 1.5 | 4.2 ± 0.9 | 3.6 ± 0.8 | 3.4 ± 1.0 |

CON, control; LEP1.0, leptin 1.0 nM; LEP3.1, leptin 3.1 nM. Data are presented as mean ± standard deviation.

## 2.4. Myocardial Infarct Size

There were no differences in myocardial infarct size between the CON and LEP3.1 groups (69 ± 7% and 71 ± 9%, respectively; $p = 0.947$). However, the addition of 1.0 nM leptin to the standard perfusion solution led to a significant reduction in infarct size (24 ± 3%; $p < 0.001$ vs. the CON and LEP3.1 groups; Figure 4).

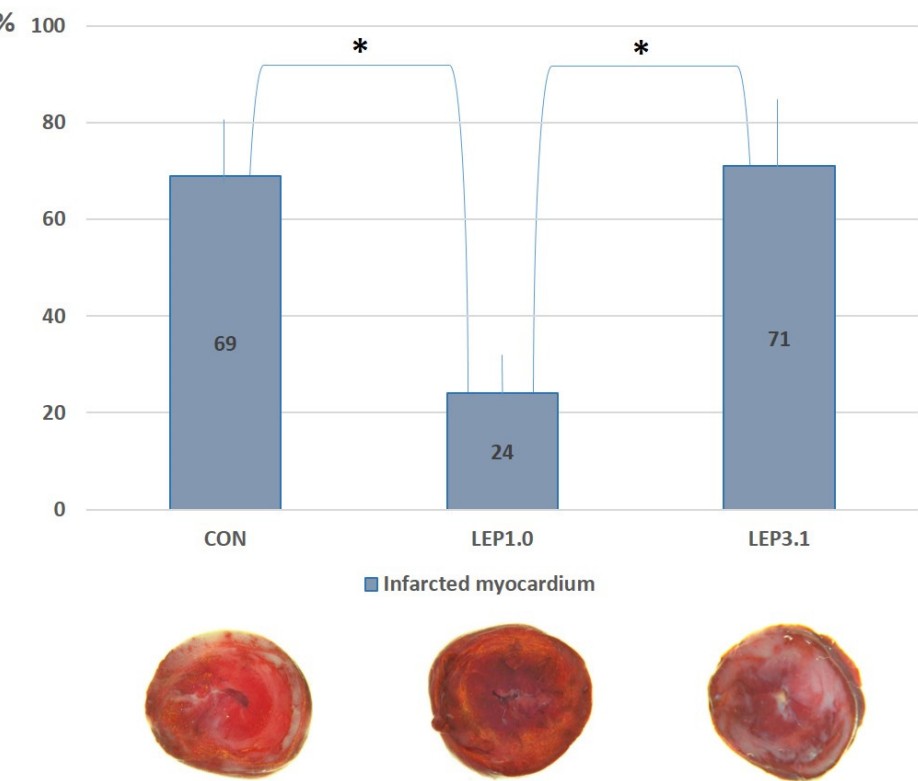

**Figure 4.** Myocardial infarct size after 120 min reperfusion of the Langendorff-perfused rat hearts subjected to 30 min of global ischemia, sectioned, and stained with 1% 2,3,5-triphenyltetrazolium chloride. CON, control; LEP1.0, leptin 1.0 nM; LEP3.1, leptin 3.1 nM. The data are presented as % of dead tissue ± standard deviation. * $p < 0.001$.

## 2.5. Rates of Ventricular Arrhythmia or Asystole during Reperfusion

The frequencies of persistent ventricular fibrillation or asystole during reperfusion were 1/10, 0/11, and 1/10 in the CON, LEP1.0, and LEP3.1 groups, respectively. Hearts with persistent ventricular fibrillation or asystole during reperfusion were not included in the analysis of post-ischemic function of the left ventricle; however, they were used for determining infarct size.

The total duration of reperfusion ventricular tachycardia was lower in the leptin groups than in the CON group, but it was significantly lower in the LEP1.0 group (Figure 5).

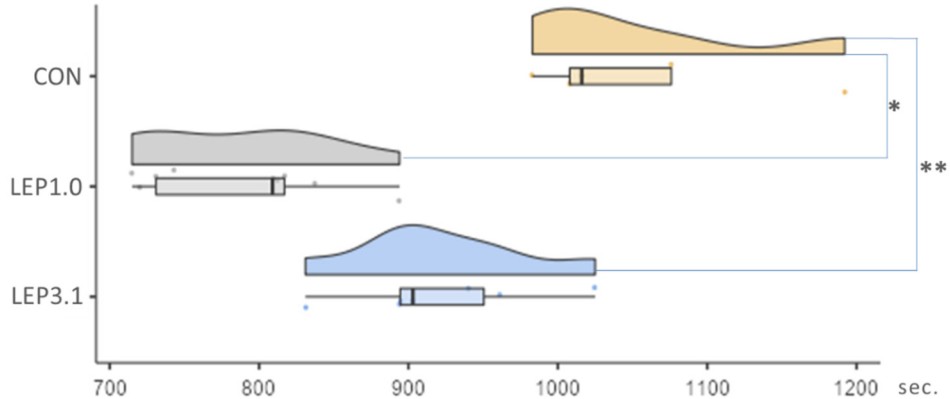

**Figure 5.** Total duration of reperfusion ventricular tachycardia. CON, control; LEP1.0, leptin 1.0 nM; LEP3.1, leptin 3.1 nM. Data are presented as mean ± standard deviation. * $p < 0.0001$; ** $p < 0.014$.

The mean HR during reperfusion-induced ventricular tachycardia was significantly lower in the LEP1.0 group than in the CON group (Figure 6).

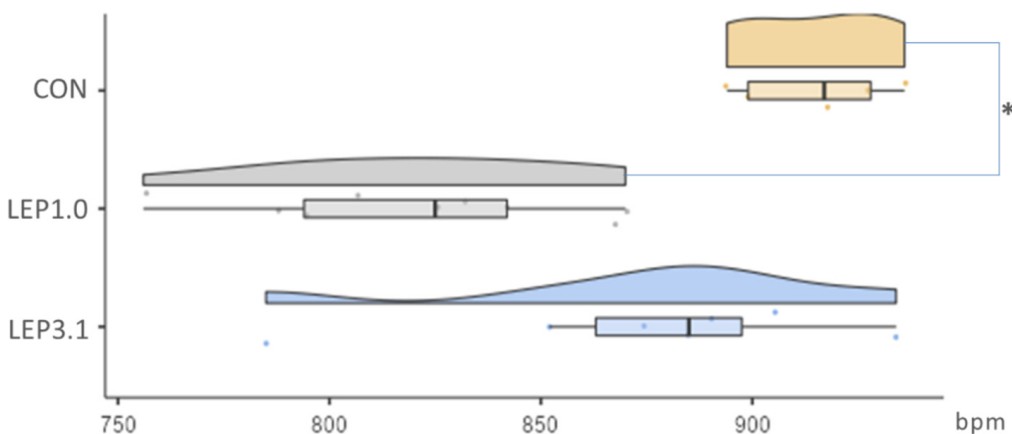

**Figure 6.** Mean heart rate (beats/min) during episodes of reperfusion-induced ventricular tachycardia. CON, control; LEP1.0, leptin 1.0 nM; LEP3.1, leptin 3.1 nM. Data are presented as mean ± standard deviation. * $p < 0.001$.

*2.6. The Expression of miRNA-144, miRNA-208a, miRNA-378 and miRNA-499 in Rat Heart after Ischemic/Reperfusion Damage*

Our results show that cardiac miRNA-208a expression is severely decreased in the LEP1.0 group compared to CON and LEP3.1 groups. Interestingly, miRNA-208a expression in the LEP3.1 group was higher than in the CON group. There was no difference in cardiac miRNA-144, miRNA-378, and miRNA-499 expression between all groups (Figure 7).

There is a negative correlation between miRNA-208a expression and the myocardial infarct size ($r = -0.748$, $p = 0.016$), suggesting that the decreased miRNA-208a expression was accompanied by attenuation of ischemic/reperfusion injury.

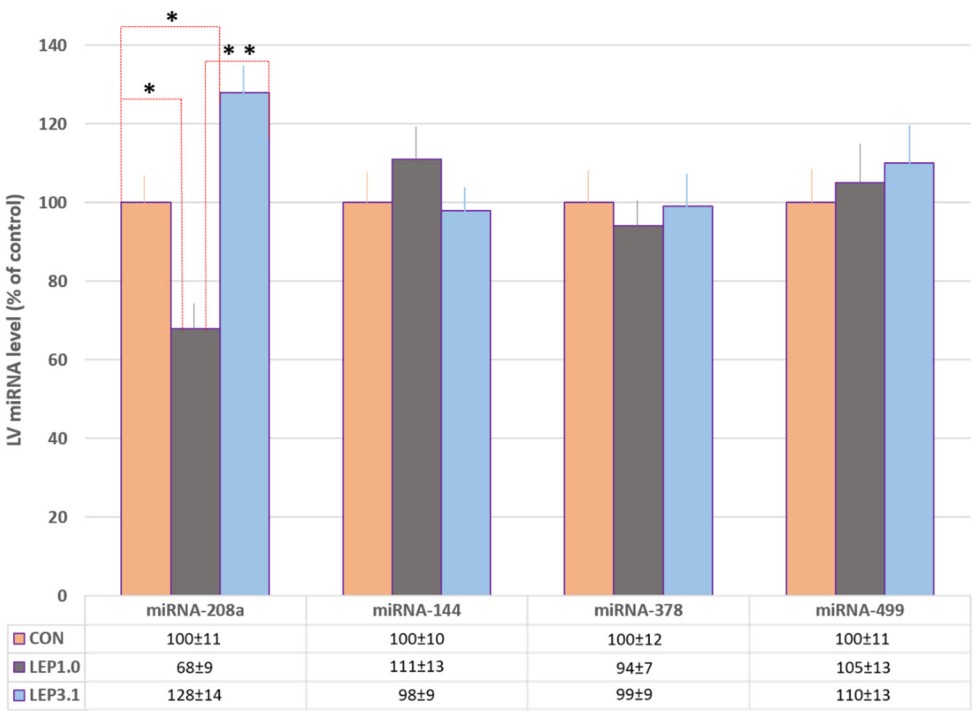

| | miRNA-208a | miRNA-144 | miRNA-378 | miRNA-499 |
|---|---|---|---|---|
| CON | 100±11 | 100±10 | 100±12 | 100±11 |
| LEP1.0 | 68±9 | 111±13 | 94±7 | 105±13 |
| LEP3.1 | 128±14 | 98±9 | 99±9 | 110±13 |

**Figure 7.** miRNA expression in rat LV myocardium. CON, control; LEP1.0, leptin 1.0 nM; LEP3.1, leptin 3.1 nM. Data are presented as mean ± standard error of the mean. * $p < 0.001$; ** $p = 0.0002$.

## 3. Discussion

The study showed that the addition of 1.0 nM leptin to the standard perfusion solution exerted a significant infarct-limiting, preserved post-ischemic ventricular function, and prevented reperfusion arrhythmia compared to the addition of 3.1 nM leptin.

Leptin may be an important factor in the regulation of cardiac function and has been associated with pathophysiological cardiovascular conditions, including coronary artery disease and congestive heart failure [3,5]. Leptin also has important effects on systemic hemodynamics and myocardial metabolism, which may also have profound effects on cardiac function [18–22].

In recent studies, the acute intraperitoneal administration of leptin was associated with a protective effect against IRI in various organs, including the brain [23], kidneys [24], gut [25], and heart [18] in rodent models. Leptin-induced cardioprotection involves activation of the reperfusion injury salvage kinase pathway through phosphatidylinositol 3-kinase cellular Akt/protein kinase B (Akt) and p44/42 mitogen-activated protein kinase, suppression of mitochondrial permeability transition pore opening, and the central leptin–proopiomelanocortin pathway, which can regulate transcription, translation, cell metabolism, and other mechanisms considered prerequisites for cardioprotection [17,26–28]. In addition, leptin signaling in the endothelium is considered protective against neointima formation in the healthy state, while obesity-induced leptin resistance can reverse this balance towards an atherogenic phenotype [29].

Leptin levels are elevated after myocardial infarction. The addition of leptin to cardiomyocytes prior to hypoxia in long-lived transgenic αMUPA mice reduces the expression of genes encoding interleukin-1-beta and tumor necrosis factor-alpha, proinflammatory cytokines implicated in non-apoptotic and apoptotic cell death [30,31]. Leptin treatment significantly reduces apoptosis in ob/ob mice and isolated myocytes; although ob/ob and db/db mice generally have normal LV systolic function, they appear to have LV diastolic functional abnormalities [32]. In the present study, the post-ischemic recovery of LVDP and LVEDP seen in the leptin group was significantly better than that in the CON group,

but leptin-mediated improvement in functional recovery was greater with the use of a low concentration of leptin.

In physiological studies in rats, leptin administration may lead to unfavorable neurohormonal changes via sympathetic nervous system activation. These effects include increased resting HR and blood pressure [33]. The resultant elevation in blood pressure may cause LV hypertrophy. However, in a mouse model of obesity, leptin had protective effects against LV hypertrophy [32]. In addition, the physiological effects of leptin include decreased appetite and increased energy expenditure, both of which can result in weight loss and improvement in hypertension, sleep apnoea, and metabolic derangements such as hyperinsulinemia and lipotoxicity [34,35]. Our study showed that the hearts in the LEP1.0 group demonstrated no ischemic contracture, and LV pressure during the entire period of global ischemia was significantly lower than that in the LEP3.1 group. This observation provides indirect support of the idea that low doses of leptin can reduce post-infarction heart failure severity [36].

Our data on Langendorff-perfused rat hearts demonstrated that adding 1.0 nM leptin to the standard KHB was accompanied by a significant reduction in infarct size, which is consistent with the findings of earlier research in rat and mouse hearts [17,18]. Recent data provided additional support for the theory that leptin administration during reperfusion may reduce cardiac IRI severity, thus providing a rationale for considering leptin as a viable therapeutic agent in the treatment of myocardial infarction [17]. However, long-term leptin treatment and high leptin concentrations have been associated with increased cellular fatty acid uptake and decreased oxidation, which over time results in intracellular lipid accumulation and lipotoxic cardiomyocyte damage and culminates in contractile failure [37]. Although myocardial metabolism was normal in the unstressed heart, cardiac-specific loss of leptin receptors completely inhibited the switch to increased glycolysis and glucose oxidation after myocardial infarction [38].

Studies on experimentally induced myocardial infarction have provided evidence of the beneficial effects of acute leptin administration. McGaffin et al. induced anterior myocardial infarction in control mice and those with cardiac-specific deletion of the leptin receptor. Mice lacking the leptin receptor in the heart developed more LV dysfunction and had a higher mortality rate after the induction of myocardial infarction [36]. The percentage of fat in the myocardium is positively correlated with body mass index and an increase in obesity [39]. Fat accumulation in the myocardium of the hearts of patients with obesity raises the question of whether leptin directly affects HR if leptin receptors are present in the sinus node and on repolarization in the ventricle. In addition to the indirect pathways through sympathetic tone, leptin can directly decrease HR and increase the QT interval through its receptors independent of β-adrenergic receptor stimulation. During the inhibition of β-adrenergic receptor activity, high concentrations of leptin in the myocardium can cause deep bradycardia, a prolonged QT interval, and ventricular arrhythmia [40]. The threshold dose of leptin to trigger norepinephrine release is approximately 1 μg/kg when introduced via an intracerebroventricular route [41]. We found that two doses (1.0 nM and 3.1 nM) of leptin did not affect the HR versus control treatment.

High concentrations of leptin can increase sympathetic activity, leading to an increased HR when the direct and indirect actions of leptin are superimposed and little change in the HR. If sympathetic signaling is impaired due to a genetic or drug effect, leptin can decrease HR [40]. The combination of direct (via leptin receptor) and indirect (via β-adrenergic receptor) effects of leptin not only provides an explanation for the slow HR and long QT in obese Zucker rats, but it also explains HR alterations in patients with obesity [42]. The results of our study showed that the total duration of reperfusion ventricular tachycardia was lower in the leptin group than in the CON group, but it was significantly lower in the LEP1.0 group. In our work, it is shown for the first time that the mean heart rate in reperfusion-induced ventricular tachycardia was significantly lower in the LEP1.0 group than in the control hearts, nevertheless, underlying mechanisms remain to be further certain. The discovery of obesity-associated molecular pathways of

cardiovascular diseases is important to improve both the prevention and management. So, miRNAs are a class of gene regulators that can bind most commonly, but not exclusively, to 3′-untranslated regions of messenger RNAs of protein-coding genes and negatively regulate their expression. MiRNAs have been shown to play a key role in cardiac structure, metabolism, and function under physiological and pathological conditions [8,9].

Some cardiac-specific miRNAs such as miRNA-144, miRNA-208a, miRNA-378, and miRNA-499 are known, however, the effect of obesity and changes in the concentration of adipose tissue hormones, in particular leptin, on cardiac miRNAs in ischemia remain obscure [43]. We revealed changes in the expression of only miRNA-208a in the myocardium depending on the concentration of leptin in rats after modeling global myocardial ischemia. MiRNA-208a regulates systemic energy homeostasis and β-myosin heavy chain content via mediator complex subunit 13 (MED13) [44]. MiRNA-208a represses MED13 inducing stress-responsive and thyroid hormone signaling pathways in the heart. MED13 acts in the heart to control cardiac remodeling as well as to increase metabolism in adipose tissue and weight loss [9,44].

We discovered that a moderate concentration of leptin (LEP1.0 group) led to a decrease in the expression of miRNA-208a, and a higher concentration of leptin (LEP3.1 group) accompanied by an increase of miRNA-208a myocardial expression. This may be one of the explanations for the obesity paradox, when moderate obesity is observed to reduce the risk of cardiovascular disease.

These findings increase our understanding of the mechanisms underlying the effects of obesity on cardiovascular function as well as the leptin effects in myocardial ischemia and suggest miRNA-208a is a potential therapeutic target for pathological conditions involving cardiac and metabolic disorders.

## 4. Materials and Methods

### 4.1. Animals

Thirty-six adult male Wistar rats (body weight, 240–300 g each) were used in the experiments. This study was conducted in accordance with the policies and procedures detailed by the 'Institutional Animal Care and Use Committee'. The animals received humane care in accordance with the European Convention on Animal Care regulations. All protocols and procedures for the animal experiments were reviewed and approved by the Committee for the Control of the Maintenance and Use of Laboratory Animals (V.A. Almazov National Medical Research Centre; 6 February 2020 application # 20-01).

### 4.2. Drugs

Recombinant rat leptin was purchased from R&D Systems (Minneapolis, MN, USA). The stock solution of leptin was prepared according to the manufacturer's recommendation and was reconstituted at 1 mg/mL in a sterile vehicle: 20 mM Tris-HCl, pH 8.0. The isolated hearts were perfused with normal buffer or buffer containing 1.0 nM [19] or 3.1 nM [20] leptin concentration.

### 4.3. Perfusion of Isolated Hearts

We followed the methods we used in our earlier work by Minasian et al. (2013) [21]. The rats were anesthetized with sodium pentobarbital (60 mg/kg, intraperitoneally). Heparin was not administered before heart excision. Each heart was excised via bilateral thoracotomy and perfused through the ascending aorta with standard Krebs–Henseleit buffer (KHB) in mmol/L: NaCl, 118.5; KCl, 4.7; NaHCO$_3$, 25; KH$_2$PO, 1.2; MgSO$_4$, 1.2; glucose, 11; and CaCl$_2$, 1.5 with or without leptin, and at a constant pressure of 85 mmHg and equilibrated with 95% O$_2$/5% CO$_2$ gas mixture at 37.0 ± 0.5 °C delivered through the inverted fritted glass filter to maintain pO$_2$ ≥500 mmHg. For the Langendorff heart preparation, the ascending aorta is cannulated and connected to a fluid reservoir, leading to retrograde flow of the medium through the aorta with steady pressure on the aortic valve at a constant pressure of 85 mmHg. Perfusion pressure was maintained by gravity, that is,

by using a water-jacketed double-walled glass column connected to the aortic cannula via a 3-way stopcock. The time interval between the opening of the thoracic cavity and initiation of perfusate flow to the heart was <80 s.

Isolated heart perfusion was performed for 60 min with KHB or KHB containing 1.0 nM or 3.1 nM leptin concentration before heart arrest and throughout the experiment. Heart function stabilized 15 minutes before the start of the experimental protocol. Left ventricular systolic (LVSP) and end-diastolic pressure (LVEDP) were measured isovolumetrically using a nonelastic polyethylene balloon introduced into the left ventricle via the left atrium. The balloon was coupled to an insulin syringe and inflated with 0.4–0.6 mL of boiled water to obtain an LVEDP <10 mmHg during stabilization. Software (PhysExp Gold, Cardioprotect Ltd., Saint Petersburg, Russian Federation) was used to process the pressure wave recorded using a miniature pressure transducer (Baxter International, Deerfield, IL, USA) and produce every minute a value corresponding to the mean LVEDP and LVSP. Left ventricular developing pressure (LVDP) was calculated as LVSP−LVEDP. Coronary flow rate (CFR) was measured by timed collection of the perfusate outflow. Heart rate was monitored using an intraventricular balloon inserted into the LV.

The standard temperature of the heart and solutions during all experiments was maintained at 37 °C. The heart temperature was maintained between standard Krebs perfusion solution by means of heart immersion into a buffer-filled water-jacketed glass chamber. The temperature in the chamber was maintained using a thermocirculator. The myocardial temperature was monitored using a miniature temperature probe inserted into the right ventricular cavity through a small incision in the pulmonary artery. We used the same carrier volume for both leptin solutions.

### 4.4. Experimental Design and Protocol

For the experiments, the rats were randomized into one of the following three groups using the random number method:

1. Control group (CON, n =12): conditioning the heart with KHB for 60 min, followed by global ischemia for 30 min and reperfusion for 120 min.

2. Leptin (1.0 nM) perfusion group (LEP1.0, n = 12): conditioning the heart with KHB + leptin (1.0 nM) for 60 min, followed by global ischemia for 30 min and reperfusion for 120 min.

3. Leptin 3.1 nM perfusion group (LEP3.1, n = 12): conditioning the heart with KHB + leptin 3.1 nM for 60 min, followed by global ischemia for 30 min and reperfusion for 120 min.

LVSP, LVEDP, LV pressure, CFR, and HR were measured at the 5th, 15th, 30th, 45th, and 60th min of KHB or KBH + leptin perfusion and at the 5th, 30th, 60th, 90th, and 120th min of reperfusion. The LV pressure was also measured at the 5th, 10th, 15th, 25th, and 30th min of global ischemia. Reperfusion rhythm disturbances were recorded according to the intraventricular pressure, while the total duration and HR were analyzed during episodes of ventricular tachycardia.

### 4.5. Exclusion Criteria

Any heart with an HR of <220 beats/min and a CFR of >18 or <8 mL/min at the end of the 15 min stabilization period was excluded from the study. Hearts failing to show an LVDP of >100 mmHg when the LVEDP was maintained at <10 mmHg were also excluded. Thus, five hearts were excluded during stabilization (two from the CON group, one from the LEP1.0 group, and two from the LEP3.1 group).

### 4.6. Infarct Size Measurement

At the end of reperfusion, the hearts were cut into four transverse slices and incubated in 1% 2,3,5-triphenyltetrazolium chloride (TTC) at 37 °C for 15 min. The TTC-stained slices were photographed with a digital camera to further determine the TTC-negative (infarcted) area. After computer planimetry (Photoshop CS, Adobe), the infarct size was expressed as

a percentage of the total ventricular area minus the cavity, and the mean value of all slices in the heart was used in the statistical analysis.

*4.7. Analysis of miRNA Expression*

Frozen LV myocardium samples were homogenized in Trizol and RNA was isolated, according to the manufacturer's instructions (Invitrogen Life Technologies, CA, USA). Then, using the NanoDrop Spectrophotometer (Nano-Drop Technologies, Wilmington, DE, USA), the total RNA concentration was quantified and checked for integrity by EtBr-agarose gel electrophoresis. Ten nanograms of the total RNA was reversely transcribed using a TaqMan miRNA reverse transcription kit (Applied Biosystems, Foster City, CA, USA) according to the manufacturer's protocol. MiRNA-144, miRNA-208a, miRNA-378, and miRNA-499, were detected using TaqMan miRNA assays (Applied Biosystems, Foster City, CA, USA). cDNA for miRNA analysis was synthesised from total RNA using miRNA-specific primers according to the TaqMan miRNA Assay protocol (Applied Biosystems, Foster City, CA, USA). The 15 μL reactions obtained by the TaqMan MiRNA Reverse Transcription Kit protocol (Applied Biosystems, California, USA) were incubated in a Thermal Cycler for 30 min at 16 °C, 30 min at 42 °C, and 5 min at 85 °C. The 20 μL PCR included 1.33 μL RT product, 10 μL TaqMan Universal PCR master mix II (2×), 7.67 μL nuclease-free water, and 1 μL of primers and probe mix of the TaqMan MiRNA Assay protocol. The reactions were incubated in a 96-well optical plate at 95 °C for 10 min, followed by 40 cycles of 95 °C for 15 s and 60 °C for 1 min. Samples were normalized by evaluating U6 expression. Relative quantities of target miRNA expressions of hyperleptinemic rats vs. control rats were compared after normalization to the values of the reference gene (ΔCT). Fold-changes in miRNA expression were calculated using the differences in ΔCT values between the samples against the mean of all control samples (ΔΔCT) and equation $2-\Delta\Delta CT$. Results were expressed as % of control. For all polymerase chain reaction experiments, samples were run in triplicates.

*4.8. Statistical Analysis*

Continuous functional data and infarct size are expressed as mean ± standard deviation (SD). The sample size per group was determined using the following parameters: SD values calculated on the basis of previous studies, desired confidence level (95%), and acceptable difference in outcome among the groups (Statistics Calculator). The statistical analyses were performed using the SPSS 20.0 software package. The Kruskal–Wallis test was used to determine the differences in infarct size. Pairwise intergroup comparisons were performed using the nonparametric Mann–Whitney U test. Differences in continuous data were tested using repeated measures analysis of variance, followed by Tukey's post hoc test. The Pearson's coefficient of correlation was used to evaluate the correlation between parametrical data. Statistical significance was set at $p < 0.05$.

**5. Conclusions**

Leptin at a concentration of 1.0nM added to the standard perfusion solution as a pharmacologic preconditioning had an infarct-limiting effect, preserved postischemic ventricular function, and prevented reperfusion arrhythmia better than 3.1nM leptin concentration. A moderate concentration of leptin (LEP1.0 group) led to a decrease in the expression of miRNA-208a, and a higher concentration of leptin (LEP3.1 group) was accompanied by an increase in miRNA-208a myocardial expression. The present study has shown that leptin is important for cardiac function and at low doses, when administered acutely, may protect against cardiac ischemia-reperfusion injury. These new data can become the basis for the pipeline of a new technology for the prevention and reduction of ischemic myocardial damage during ischemia and reperfusion.

## 6. Study limitations

This paper revealed some molecular facts about the transcriptional regulation of the miRNA208a response to leptin. However, further experimental evidence is needed to further support this idea about posttranscriptional regulations of gene expression and the downstream targets of miRNA 208a, including MED13, was a potential target.

**Author Contributions:** Conceptualization, E.A.P. and M.M.G.; methodology, E.A.P. and M.M.G.; software, E.A.P.; validation, E.A.P., M.M.G. and S.M.M.; formal analysis, S.M.M.; investigation, E.A.P. and S.M.M.; resources, E.A.P. and M.M.G.; data curation, M.M.G. and E.N.M.; writing—original draft preparation, E.A.P.; writing—review and editing, E.A.P. and M.M.G.; visualization, E.A.P. and S.M.M.; supervision, M.M.G. and E.N.M.; project administration, M.M.G. and E.V.S.; funding acquisition, M.M.G. and E.V.S. All authors have read and agreed to the published version of the manuscript.

**Funding:** The study has been supported by the grant from the Ministry of Science and Higher Education of the Russian Federation (agreement 075-15-2020-800).

**Institutional Review Board Statement:** All animal experimental protocols and procedures were reviewed and approved by the Committee for the Control of the Maintenance and Use of Laboratory Animals of the V. A. Almazov National Medical Research Center (date: 06-FEB-2020, application number: 20-01).

**Informed Consent Statement:** Not applicable.

**Data Availability Statement:** The data presented in this study are available on request from the corresponding author.

**Conflicts of Interest:** No authors have any conflict of interest during the publication of the manuscript.

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
