# Peer review of "Anti-Ischemic Effect of Leptin in the Isolated Rat Heart Subjected to Global Ischemia-Reperfusion: Role of Cardiac-Specific miRNAs"

_cardiogenetics, doi:10.3390/cardiogenetics13010001_

Round 1
Reviewer 1 Report
I have been asked to review this manuscript on the role of cardiac-specific miRNAs of anti-ischemic effect of leptin in the isolated rat heart subjected to ischemia-reperfusion.
The role of leptin has been previously well reported in myocardial ischemic reperfusion injuries.
The introduction is rather long. It can be shortened to focus on the study. Introduction, may include the reasons for selecting the particular miRNAs.
The miRNA-208a expression was lower in the low dose leptin group than in controls and this was associated with better hemodynamics (LVDP and LVEDP) and small infarct in the low dose leptin group. There is however a paradox in the high dose leptin group. Though the miRNA-208a expression was increased in the high dose leptin group, the hemodynamics were worse that that of the controls. Does it mean that the miRNA-208a expression did not correlate with cardiac function. Similarly, the infarct size was not increased the high dose leptin group despite higher levels of miRNA-208a expression. This indicates that the infarct size did not correlate with miRNA-208a expression in these groups. The results thus have some contradictions. The authors need to discuss and clarify this.
Similarly, the discussion should also focus on the current study that is ischemia reperfusion injury. Parts on heart failure may be omitted.
The conclusion is too emphatic. Authors have only shown a decrease in miRNA-208a myocardial expression in the low dose leptin group. They haven’t studied the downstream pathway and the fact that miRNA-208a expression may be one the reasons for the obesity paradox cannot be concluded. This should be removed from the conclusions but may be brought out in the discussion.
Author Response
Dear reviewer, thank you for the detailed analysis of the article, recommendations and comments.
All comments have been corrected. The introduction has been shortened and supplemented where necessary. Parts on heart failure were omitted. The conclusion was corrected.
Reviewer 2 Report
This paper entitled “anti-ischemic effect of leptin in the isolated rat heart subjected to global ischemia-reperfusion: role of cardiac-specific miRNAs” described the affection of dosage dependent of leptin on ischemia-reperfusion injury. The authors found that a low concentration of leptin (1nM), but not a high concentration (3nM) could preserve post ischemic ventricular function and prevent reperfusion arrhythmia. They further investigated the role of miRNA in this process and found that the transcription of miRNA208 is associated with leptin concentration.
Overall, this is an interesting paper with important findings about the relationship between leptin and cardiac function. The experimental design and data presentation are overall acceptable, but a few missing points were missing to link to leptin manipulation and molecular mechanisms, which are listed below. Additional experiments will be required:
1. The main aim of this paper is to identify the role of miRNA in ischemia-reperfusion when apply leptin. More information is needed in the introduction session to introduce the known function of cardiac specific miRNA, and why some specific miRNAs were selected in this study.
2. This paper revealed some molecular facts about the transcriptional regulation of the miRNA208a response to leptin. However, further experimental evidence is needed to further support this idea. How is miRNA 208a’s transcription activated by leptin? What are the downstream targets of miRNA 208a? The authors hypothesized MED13 was a potential target, but further evidence is required to link the leptin response to the function of MED13 via miRNA208a. Will be the MED13 transcription triggered by leptin? Also, will heart rate and other factor changes by disruption of MED13? Filling these gaps will significantly complete this study.
3. A Few pieces of important information is missing:
3.1 line 79: initials (e.g., LVDP, LVEDP) need to be fully explained when the first time shown in the paper. This also applies for Line 84 (CON), line 118(CFR, HR).
Author Response
Dear reviewer, thank you for the detailed analysis of the article, recommendations and comments.
All comments have been corrected. Several important details have been added.
The limitations of the study are described.
Reviewer 3 Report
This is a very interesting study regarding the anti-ischemic effect of leptin in rats. The authors demonstrated that leptin is important for cardiac function and, administered acutely, at low doses, it may protect against cardiac-ischemia-reperfusion injury. I consider that the article is interesting and may have important implications for clinical practice.
1. I consider it is important to present more details about the inclusion and exclusion criteria in the study.
2. Please include the details of the broader impacts of the study made, addressing the future scope and topics that are important.
3. Please consider including the limitations of the study.
Author Response

(The authors gave the same response as above.)

Round 2
Reviewer 2 Report
I'm pleased to see all my concerns have been addressed and have no more questions prior to its publication.